**Subject Category:**
Biology (whole organism)

ecology/evolution/psychology

body size, costly signalling, trophy hunting, wildlife harvest, exploitation, carnivore

**Author for correspondence:**
Ilona Mihalik
e-mail: ilonammillie@gmail.com

# Trophy hunters pay more to target larger-bodied carnivores

Ilona Mihalik[1,2], Andrew W. Bateman[1,2]
and Chris T. Darimont[1,2]

[1]Department of Geography, University of Victoria, David Turpin Building, 3800 Finnerty Road, Victoria, British Columbia, Canada V8P 5C2
[2]Raincoast Conservation Foundation, Sidney, British Columbia, Canada V8L 2P6

IM, 0000-0003-4829-7598; CTD, 0000-0002-2799-6894

Hunters often target species that require resource investment disproportionate to associated nutritional rewards. Costly signalling theory provides a potential explanation, proposing that hunters target species that impose high costs (e.g. higher failure and injury risks, lower consumptive returns) because it signals an ability to absorb costly behaviour. If costly signalling is relevant to contemporary 'big game' hunters, we would expect hunters to pay higher prices to hunt taxa with higher perceived costs. Accordingly, we hypothesized that hunt prices would be higher for taxa that are larger-bodied, rarer, carnivorous, or described as dangerous or difficult to hunt. In a dataset on 721 guided hunts for 15 North American large mammals, prices listed online increased with body size in carnivores (from approximately $550 to $1800 USD/day across the observed range). This pattern suggests that elements of costly signals may persist among contemporary non-subsistence hunters. Persistence might simply relate to deception, given that signal honesty and fitness benefits are unlikely in such different conditions compared with ancestral environments in which hunting behaviour evolved. If larger-bodied carnivores are generally more desirable to hunters, then conservation and management strategies should consider not only the ecology of the hunted but also the motivations of hunters.

## 1. Introduction

The behaviour of human hunters and fishers diverges substantially from other predators of vertebrate prey. Instead of targeting mainly juvenile or otherwise vulnerable individuals, humans (often males) typically seek large taxa, as well as large, reproductive-aged individuals within populations [1–5], targets also sought by early human groups [6]. This distinct pattern of hunting behaviour is likely shaped by multiple selective forces

[7]; for example, in subsistence societies, targeting large prey items may be motivated by kin provisioning [8–11], whereas widely sharing large prey beyond kin, and expecting the same in return, may follow reciprocal altruism [12,13].

Additional patterns have informed other evolutionary explanations underlying hunting behaviour. Within traditional hunter–gatherer groups, for example, male hunters often target species with a highly variable caloric payoff over more reliably or safely acquired alternatives [14]. Particularly in trophy hunting contexts, contemporary hunters often similarly pursue taxa that are rare [15–19]. Additionally, owing to restrictions on meat exports, and to the targeting of seldom-eaten species, such as large carnivores, professionally guided hunters frequently seek prey without the intention of receiving nutrition, the primary benefit of predation in the wild. Such seemingly inefficient behaviour begs the questions: how did such behaviour evolve, and why might it persist today?

Ostensibly wasteful investments by animals have long intrigued researchers, inspiring theory, empirical investigation and debate. Darwin [20], for example, questioned what drove the evolution of extravagant traits in males, such as the large tails of peacocks (*Pavo* spp.) and antlers of deer (Cervidae). Zahavi [21] proposed that time-consuming, risky, inefficient or otherwise 'handicapping' traits or activities could be interpreted as 'costly signals'. Costly signalling theory suggests that a costly signal reflects the capacity of the signaller to bear the cost, thereby providing honest information to potential mates and competitors about the underlying quality of the signaller [21] (e.g. the 'strategic cost' [22]). The theory suggests that honesty is maintained through the differential costs and benefits of signal production; individuals of higher quality are thought to better afford the larger costs associated with more attractive signals, while the costs outweigh the benefits and signals are difficult to fake for lower-quality individuals [22–24]. Under this framework, evolutionary benefits flow to higher-quality signallers as well as signal recipients. For example, in avian courtship displays, male birds subject themselves to predation risk by singing or dancing in the open during sexual displays, signalling that they have underlying qualities that permit them to absorb the energetic and predation-risk costs of the display [21]. In human systems, costly signalling has been used to explain behaviour associated with artistic elaboration, ceremonial feasting, body modification and monumental architecture [5,25]. Individuals that can afford costly signals can attract mates or accrue social status, which can increase access to resources (e.g. foods, material goods, approval from peers, knowledge) [21,26].

Costly signalling has also been invoked to explain hunting behaviour in some human subsistence systems, although relevant data are limited and debate is common [10,27–29]. According to the theory in this context, when subsistence hunters target items with high costs, they honestly signal their ability to absorb the costs [14,30]. Thus, hunting itself serves as the signal, and successfully hunting a species with high costs signals higher quality (akin to a more showy avian courtship display). Hunting of marine turtles (*Chelonia mydas*) by the Meriam peoples of Murray Island, Northern Australia, provides an example. There, diverse members of Meriam society collect marine turtles as they crawl on the beach where they are easily captured; however, only reproductive-aged men participate in offshore turtle hunting, a costly activity (i.e. high risk of failure; increased risk of injury; lower consumptive returns; high energetic, monetary, time investment costs) [25,31,32]. When successful, these hunters rarely consume the meat themselves, and instead provision community members at large feasts, arguably providing the public forum to signal the hunters' underlying qualities that allow them to engage in such costly behaviour [25,31,32]. Successful Meriam turtle hunters earn social status and higher reproductive success, providing rare evidence for fitness benefits associated with apparent costly signalling in humans [31,32]. Men from other hunter–gatherer societies suggested to exhibit similar signalling behaviour, not easily explained by provisioning or reciprocal altruism alone, include the Ache men of Eastern Paraguay [30], the Hadza men of Tanzania [33] and male torch fishers of Ifaluk atoll [34]. However, some criticisms of these interpretations include whether men's hunting patterns are truly suboptimal in terms of nutrient acquisition (e.g. argued in the case of the Hadza men [27]) and that Hadza [28] and Ache [29] men value provisioning over showing-off their hunting ability, regardless of having dependent offspring. Others argue that fitness benefits gained by hunters are influenced by multiple pathways, rather than just through showing off [10].

Although a controversial theory when applied to human subsistence-hunting, examining seemingly wasteful hunting behaviour among non-subsistence hunters (hunting without the goal of providing food, e.g. trophy hunting) offers new opportunities to confront elements of costly signalling. In particular, non-subsistence hunters appear to incur substantial costs—in terms of high failure risk or risk of injury, as well as low to nil consumptive returns—when they target large-bodied, carnivorous, rare and/or dangerous or difficult-to-hunt species. Specifically, we would expect increased failure risk via lower encounter rates with larger and higher trophic-level animals, which tend to occur at lower densities

**Table 1.** North American 'big game' species included in our study.

| species (common) | Latin | classification |
|---|---|---|
| mountain lion | *Puma concolor* | carnivore |
| black bear | *Ursus americanus* | carnivore |
| brown bear | *Ursus arctos* | carnivore |
| polar bear | *Ursus maritimus* | carnivore |
| muskox | *Ovibos moschatus* | ungulate |
| gray wolf | *Canis lupus* | carnivore |
| thinhorn sheep | *Ovis dalli* | ungulate |
| bighorn sheep | *Ovis canadensis* | ungulate |
| caribou | *Rangifer tarandus* | ungulate |
| pronghorn | *Antilocapra americana* | ungulate |
| white-tailed deer | *Odocoileus virginianus* | ungulate |
| moose | *Alces alces* | ungulate |
| mule deer | *Odocoileus hemionus* | ungulate |
| mountain goat | *Oreamnos americanus* | ungulate |
| elk | *Cervus canadensis* | ungulate |

than small, low-trophic-level species [35]. Similarly, hunters likely encounter other rare species less frequently than abundant species. In addition, species that are dangerous or difficult to hunt are likely to increase failure and injury risk, posing another cost. Moreover, hunters often kill seldom-eaten species, such as carnivores, which includes the opportunity cost of forgoing greater nutrition from hunting edible prey. Collectively, hunting inefficiently by targeting such prey could signal a perceived ability to accept the costs of higher failure and injury risk, as well as opportunity costs, compared with targeting species that are more easily secured and offer a higher nutritional return. Throughout this paper, we use the term 'cost' to refer to these opportunity costs (lower nutritional returns) as well as failure and injury risks; by contrast, we use the term 'price' (see below) when referring to the money hunters pay for guided hunts.

Although the targeting of some big game (i.e. large mammals hunted for sport) by modern non-subsistence hunters appears to include elements of costly signalling behaviour, there have been no empirical evaluations of the theory in this context. If such behaviour persists among contemporary hunters, we would predict that species with high perceived costs should be more desirable to hunters because they could signal a greater ability to absorb the costs. Accordingly, assuming that market demand influences price to reflect desirability—a common assumption [15–19]—we hypothesized that hunt prices would be higher for taxa with higher perceived costs of hunting. We note that lower supply, through rarity or hunting restrictions, could also drive up prices, but we would not expect to find an association with prey body size, hunt danger or difficulty in this case. We confronted our hypothesis using data from guided trophy hunting systems, where hunters hire specialist guides [36]. Prices for guided hunts can be substantial, ranging from several hundred to many thousands of US dollars (USD) per day [15–17]. Specifically, using price charged per day for guided hunts as an index, we predicted that species that are (1) large-bodied, (2) rare, (3) carnivorous and (4) described by Safari Club International (SCI) [37] as dangerous or difficult to hunt would be priced higher.

# 2. Data collection and methods

## 2.1. Species included

We collected prices advertised for guided hunts of 15 North American big game species (table 1). We selected these species because they comprise the species requirements of SCI's *Grand Slam North American 29* award, which requires hunters to kill 29 of the 38 North American species or subspecies [38].

## 2.2. Price data

We collected data on prices advertised online by hunting guides for each species in every North American province and state in which species occurred ($n = 160$ species-jurisdiction combinations). Data collection occurred between November 2017 and January 2018. We used consistent web searches using the Google search engine, changing only the jurisdiction or species name for each new search. The goal was to obtain the first five unique prices from each search. Forty-six of the 160 search combinations, however, revealed fewer than five results (mean = 2.37, range: 1–4). Owing to particularly low sample size for polar bears (originally, two), we collected four additional polar bear prices in June 2018. For these, we used the term 'arctic' in place of the jurisdiction and altered the order of wording. From all price data, we calculated an average price/day.

Websites presented a variety of options to hunters, requiring a standardization approach. We excluded websites that either did not include prices or stated to 'call or email for prices'. We focused on free-range (i.e. not fenced) hunts targeting males during the rut (for ungulates) if different seasons were available. Prices included guiding, meals and accommodation. We did not include baited hunts or 'combination' hunts that included more than one species for one price. We chose rifle hunts when hunters were given options (usually among rifle, muzzleloader, archery), and muzzleloader was chosen if the rifle was not an option. If neither were available ($n = 3$), archery hunt prices were converted to firearm prices by using the average ratio (rifle price/archery price; 1.20) calculated from those with a choice ($n = 26$).

We estimated the contribution of charter flights to the total cost to remove that component from prices that included it ($n = 49$). We subtracted the average flight cost if included, calculated from hunts that stated the cost of a charter for the same species-jurisdiction. If no estimates were available, the average flight cost was estimated from other species within the same jurisdiction, or from the closest neighbouring jurisdiction. Similarly, trophy and licence/tag fees (set by governments in each province and state) were removed from prices if they were advertised to be included.

We also estimated a price-per-day from hunts that did not advertise the length of the hunt. We used data from websites that offered a choice in the length (i.e. 3 days for $1000, 5 days for $2000, 7 days for $5000) and selected the most common hunt-length from other hunts within the same jurisdiction. We used an imputed mean for prices that did not state the number of days, calculated from the mean hunt-length for that species and jurisdiction.

Overall, we obtained 721 prices for 43 jurisdictions from 471 guide businesses. Most prices were listed in USD, including those in Canada. Ten Canadian results did not state the currency and were assumed as USD. We converted CAD results to USD using the conversion rate for 15 November 2017 (0.78318 USD per CAD).

## 2.3. Body mass

Mean male body masses for each species were collected using three sources [37,39,40]. When mass data were only available at the subspecies-level (e.g. elk, bighorn sheep), we used the median value across subspecies to calculate species-level masses.

## 2.4. Rarity

We used the provincial or state-level conservation status (the subnational rank or 'S-Rank') for each species as a measure of rarity. These were collected from the NatureServe Explorer [41]. Conservation statuses range from S1 (Critically Imperilled) to S5 (Stable) and are based on species abundance, distribution, population trends and threats [41]. Some ranks denote uncertainty within a range and fall in-between two ranks (e.g. S1S2; S2S3, etc.). Accordingly, we converted ranks to 1, 1.5, 2, 2.5, 3, 3.5, 4, 4.5, 5.

## 2.5. Classification: ungulate versus carnivore

We categorized species as either 'carnivore' or 'ungulate'. Carnivore includes species of the order *Carnivora* and ungulate includes species with hooves (table 1).

## 2.6. Difficult or dangerous

Whereas larger, rarer and carnivorous animals would carry higher costs owing to lower densities, we additionally considered other species characteristics that would increase cost due to risk of failure or potential injury. Accordingly, we categorized hunts for their perceived difficulty or danger. We scored

this variable by inspecting the 'remarks' sections within SCI's online record book [37], similar to the qualitative exploration of SCI remarks by Johnson *et al.* [16]. Specifically, species hunts described as 'difficult', 'tough', 'dangerous', 'demanding', etc. were noted. Species with no hunt descriptions or described as being 'easy', 'not difficult', 'not dangerous', etc. were scored as not risky. SCI record book entries are often described at a subspecies-level with some subspecies described as difficult or dangerous and others not, particularly for elk and mule deer subspecies. Using the subspecies range maps in the SCI record book [37], we categorized species hunts as presence or absence of perceived difficulty or danger only in the jurisdictions present within the subspecies range.

## 2.7. Statistical methods

We employed information-theoretic model selection using Akaike's information criterion (AIC) [42] to gauge support for different hypotheses relating our selected predictors to hunting prices. In general terms, AIC rewards model fit and penalizes model complexity, to provide an estimate of model performance and parsimony [43]. Before fitting any models, we constructed an *a priori* set of candidate models, each representing a plausible combination of our original hypotheses (see Introduction).

Our candidate set included models with various combinations of our potential predictor variables as main effects. We did not include all possible combinations of main effects and their interactions, and instead evaluated only those that expressed our hypotheses. We did not include models with (ungulate versus carnivore) classification as a term on its own. Given that some carnivore species are commonly perceived as pests (e.g. wolves) and some ungulate species are highly prized (e.g. mountain sheep), we did not expect a stand-alone effect of classification. We did consider the possibility that mass could influence the response differently for different classifications, allowing for an interaction between classification and mass. Following similar logic, we considered an interaction between SCI descriptions and mass. We did not include models containing interactions with conservation status as we predicted rare species to be expensive regardless of other characteristics. Similarly, we did not include models containing interactions between SCI descriptions and classification; we assumed that species described as difficult or dangerous would be more expensive regardless of their classification as carnivore or ungulate.

We fit generalized linear mixed-effects models, assuming a gamma distribution with a log link function. All models included jurisdiction and species as crossed random effects on the intercept. We standardized each continuous predictor (mass and conservation status) by subtracting its mean and dividing by its standard deviation. We fit models with the *lme4* package version 1.1–21 [44] in the statistical software R [45]. For models that encountered fitting problems using default settings in *lme4*, we specified the use of the *nlminb* optimization method within the *optimx* optimizer [46], or the *bobyqa* optimizer [47] with 100 000 set as the maximum number of function evaluations.

# 3. Results

We compared models including combinations of our four predictor variables to determine if prey with higher perceived costs were more desirable to hunt, using price as an indication of desirability. Our results suggest that hunters pay higher prices to hunt species with certain 'costly' characteristics, but do not provide support for all our hypotheses. Our highest-ranking model (table 2) included mass, classification and their interaction (figure 1). The interaction term between mass and classification (coefficient estimate = 0.38, 95% CI 0.08–0.69) indicated that hunt price increased with body mass in carnivore species but was not influenced by changing ungulate body mass. Model predictions show that prices charged to hunt carnivore species increased from approximately \$550 USD/day to \$1800 USD/day across the variation in body mass, while prices remained at approximately \$900 USD/day for ungulate species regardless of their body size (figure 1).

Although the highest-ranking model did not have overwhelming support (Akaike model weight, AIC$w$ = 0.23), the trend of higher prices for larger carnivores was consistent throughout our results, including two other models within the top-model set ($\Delta$AIC < 2). Owing to the recent debate in the literature regarding methods to determine the importance of multiple processes [48–50], we also present results in the electronic supplementary material from the model-averaged and global models. Results were qualitatively unchanged (electronic supplementary material, figure S1a,b).

SCI descriptions and rarity were not included within the top model and did not have a meaningful influence on price. SCI was included in the third-ranking model ($\Delta$AIC = 1.54, AIC$w$ = 0.11); however, the confidence interval for the effect overlapped zero (coefficient estimate = 0.06, 95% CI −0.12 to 0.24).

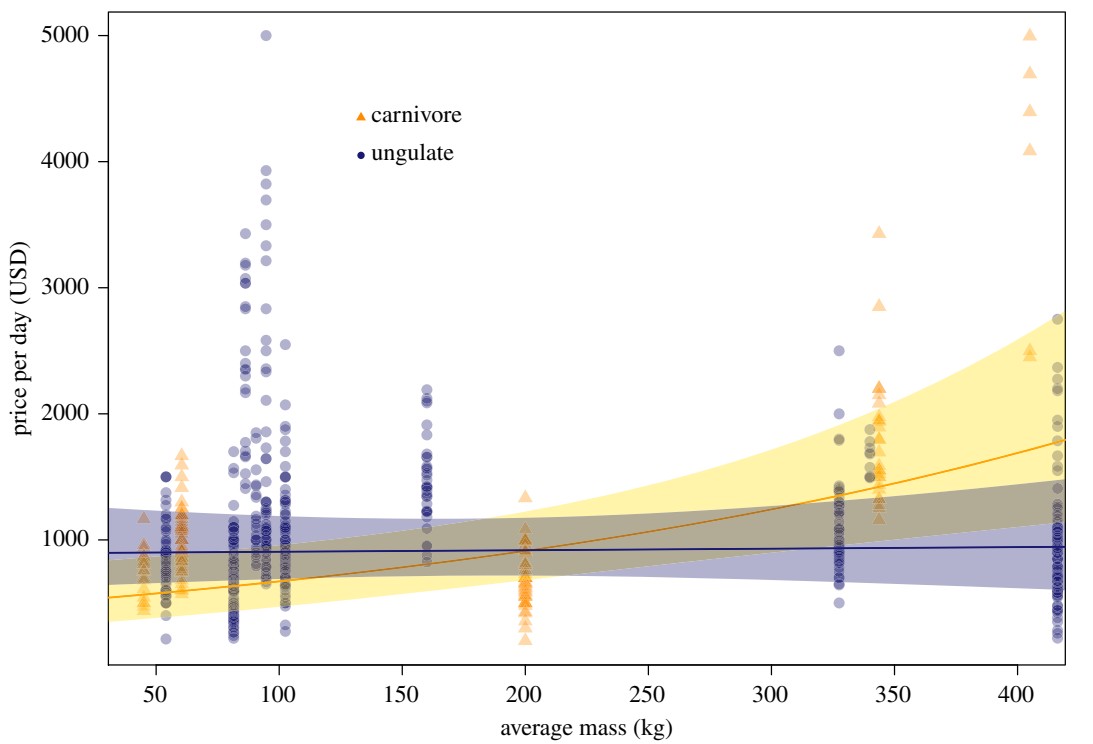

**Figure 1.** Effect of mass on the daily guided-hunt price for carnivore (orange) and ungulate (blue) species in North America. Points show raw mass for carnivores and ungulates, curves show predicted means from the maximum-parsimony model (see text) and shading indicates 95% confidence intervals for model-predicted means.

**Table 2.** AIC evaluation of models for predicting prices of hunts offered by hunting guides. The explanatory variables were the average male body mass (mass; in kg), the classification (class; either carnivore or ungulate), descriptions of difficulty or danger in Safari Club International hunt descriptions (SCI; either absence or presence) and conservation status in jurisdiction (status; 1, 1.5, 2, …, 5).

| model | d.f. | logLik | $\Delta$AIC | AICw |
|---|---|---|---|---|
| mass + class + mass × class | 7 | −4877.68 | 0.00 | 0.23 |
| mass | 5 | −4880.25 | 1.13 | 0.13 |
| class + mass + SCI + mass × class | 8 | −4877.45 | 1.54 | 0.11 |
| class + mass + status + mass × class | 8 | −4877.50 | 1.64 | 0.10 |
| SCI + mass + SCI × mass | 7 | −4878.71 | 2.05 | 0.08 |
| status + mass | 6 | −4880.09 | 2.81 | 0.06 |
| class + mass + SCI + mass × class + SCI × mass | 9 | −4877.14 | 2.92 | 0.05 |
| SCI + mass | 6 | −4880.16 | 2.95 | 0.05 |
| null (intercept only) | 4 | −4882.32 | 3.27 | 0.05 |
| SCI + mass + status + SCI × mass | 8 | −4878.59 | 3.82 | 0.04 |
| status + class + SCI + mass + SCI × mass + mass × class | 10 | −4877.00 | 4.64 | 0.02 |
| SCI + mass + status | 7 | −4880.01 | 4.67 | 0.02 |
| status | 5 | −4882.14 | 4.92 | 0.02 |
| SCI | 5 | −4882.29 | 5.23 | 0.02 |
| SCI + status | 6 | −4882.13 | 6.90 | 0.01 |

Rarity, based on species' jurisdiction-level conservation statuses, was included in the fourth-ranked model ($\Delta$AIC = 1.64, AICw = 0.10), and its confidence interval also overlapped zero (coefficient estimate = 0.01, 95% CI −0.03 to 0.05). Although both predictors did appear in the top-model set, their

effects were minimal in the model-averaged and global model results (electronic supplementary material, figures S2a,b and S3a,b).

# 4. Discussion

At a North American continental scale, we analysed guided trophy hunting in the context of costly signalling theory. We examined hunting as a signal, and the risks of failure and injury, as well as opportunity costs related to low consumptive returns, as the potential associated costs. We asked if characteristics of prey associated with higher perceived costs were correlated with higher prices charged to hunters (which we assume to represent a market-mediated index of desirability). We argue that costly signalling theory could provide an evolutionary explanation for why big game hunters target specific species [7]. We found some support for our prediction, showing that hunters pay more to kill larger-bodied carnivores, which likely carry the higher perceived risk of failure and injury, as well as low consumptive returns.

Some patterns we observed differed from previously published findings. For one, the jurisdiction-level conservation status (state or provincial-level within North America) of a species (our proxy for rarity) did not affect price in our analysis. By contrast, larger-scale conservation rankings (such as IUCN and/or CITES) have previously been found to correlate with hunting price in Caprinae species [15], Bovidae taxa [16,17], ungulates [18] and African felids [19]. Two explanations for why we did not detect a relationship might be relevant. First, the jurisdiction-level rankings showed little variation across species in our dataset; only 18% ($n = 27$) of the species-jurisdiction combinations ranged from 'Vulnerable'/'Apparently Secure' (S3/S4) to 'Critically Imperiled' (S1). The remainder was not classified as at risk. Second, hunters (especially when they commonly originate from afar) might not know or consider the jurisdiction-level conservation status (which were included in our models) of their targets, whereas a species' IUCN ranking is typically well-known and often included in their SCI record book descriptions. However, only two species used in our study were ranked by the IUCN as 'Vulnerable', while the remaining 13 ranked as 'Least Concern'.

We found that the presence of a 'difficult and/or dangerous' hunt description by SCI [37] likewise had no statistical influence on price. This result departed from our predictions, given that difficult and dangerous descriptions should increase the perception of failure risk and risk of injury. We speculate that, unlike subsistence hunts (which likely carry a realistic and meaningful risk of failure), guided big game hunters in reality risk relatively little in terms of failure owing to difficulty or danger. Contemporary hunters now employ efficient killing technology to hunt prey at a safe distance [36,51]. Indeed, while we expected the perception of difficulty and danger to matter in terms of desirability, guided hunts that pose real risks to safety might be relatively uncommon, and guided clients are likely to be aware of this.

Our work has several potential limitations. Among them, we assume that prices charged to hunt different species reflect desirability for hunters, an assumption commonly made in related literature [15–19]. Additional factors are likely also involved. While we did not address it in our study, due to the coarse state- or province-scale resolution of available data, the cost of living (food, accommodation and guiding) may also influence prices. Given that the two largest carnivores (polar and grizzly bears) in our dataset occur at northern latitudes, associated with remoteness and high costs of living, this was of concern. Accordingly, we examined *post hoc* whether latitude could explain the high hunt prices observed for large carnivores. While large carnivores do tend to occur at higher latitudes (electronic supplementary material, figure S4), we found no statistical evidence that latitude drove hunt price for carnivores (electronic supplementary material, figure S5). Additionally, some might argue that pursuing larger-bodied carnivores might have additional costs related to searching for targets, given their naturally low density. This is possible, but we standardized our price metric to daily rates, dealing with the possibility that lower density species might take longer to locate. Furthermore, the use of an imputed mean for hunts without a listed duration, calculated by using the mean hunt-length for a species-jurisdiction (combination of each species in every North American province and state in which they occur), could lead to biased results for carnivores if they do indeed require additional search times. Finally, we acknowledge Google's search results may vary across users and limit reproducibility [52].

We argue that the relationship between body mass and price is evident only in carnivores (figure 1) because larger size carnivores strongly signal increased danger or rarity. Specifically, although not

captured in SCI descriptions, larger-bodied carnivores could give the perception of increased danger; displaying a carcass of a predator could signal the absorbed costs of interacting with animals that, compared to ungulates, are perceived as more dangerous if they are larger-bodied. Additionally, larger-bodied carnivores are naturally rarer, owing to their higher trophic position [35]. This dimension of rarity (*perceived rarity* [53]) could be recognized by hunters and could therefore serve as a better proxy for rarity than conservation status, especially on a continent where few hunted taxa are of conservation concern. Finally, unlike herbivores, carnivores are generally not consumed, imposing the additional cost of receiving no nutritional gains from kills. Only the smaller-bodied black bear (classified here as a carnivore) is commonly eaten. While these explanations are speculative, they generally align with previous research that has found North American hunters display evidence of 'achievement satisfaction' (congruence of goals and outcomes regarding performance) more commonly when sharing information about carnivore hunts compared to herbivore hunts. For example, men posing with carnivores of any size in hunting photographs have higher odds of displaying a 'true smile', an honest signal of pleasure, compared to pictures with herbivore prey [54]. Additionally, in online discussion forums about hunting, men express achievement-oriented phrases more frequently when describing carnivore hunts compared to ungulate hunts [55].

Our results, showing the increased value placed by hunters on large-bodied prey, share similarities with work conducted in other areas that adopted a different line of conceptual inquiry. Specifically, the anthropogenic Allee effect (AAE) describes a phenomenon in which rare species become more desirable to hunters [15]. In this context, others have similarly found that body size positively correlates with hunting prices, specifically in ungulates [18] and African species [16]. Our results thus increase the scope of taxa and contexts involved in the pattern, suggesting that, although not universal, the desire of hunters to kill larger species exists across different environments, cultures, conservation contexts and communities of species available for hunting. This observation of similar patterns across diverse systems of contemporary hunting suggests the potential for an underlying evolutionary origin of the behaviours involved.

Costly signalling and associated theory provides a useful framework with which to evaluate the evolution and persistence of apparently inefficient behaviour in trophy hunting systems, but care in use and interpretation is required. The theory is argued by some to have been misapplied in studies of contemporary human behaviour [56]. Given that our work only relates to one prediction within the framework (that hunters should be willing to pay more to hunt species perceived as imposing higher costs), further work is required to elucidate the potential relevance of the theory in this context. We did not evaluate any fitness benefits of costly signalling to guided hunters, for example, but such benefits seem unlikely. Persistence of evolutionarily mismatched behaviours, however, is common in contemporary human society (e.g. gambling [57], risk-taking in adolescents [58]) and seems likely in this case, given differences between current social and ecological environments and the ancestral environments in which hunting behaviour evolved. However, elaborate awards from, and status hierarchies within, organizations with large followings (e.g. SCI) provide evidence of modern-day social benefits to signallers. Although there is general societal disapproval for trophy hunting, SCI offers dozens of awards that create status hierarchies among members; for example, to achieve the *World Hunting Award*, one must have already achieved 11 *Grand Slam Awards*, 17 diamond-level *Inner Circle Awards*, and both the *Fourth Pinnacle of Achievement* and *Crowning Achievement Award* [38]. Future studies could assess the relationships between costs absorbed and measures of related social status earned; with an online and increasingly globalized audience, examinations of the support (e.g. 'likes' or other positive feedback received on social media platforms) in big game hunting contexts could yield new insight. Work is also required to examine the potential benefits flowing to signal recipients, asking what information on signaller quality might be assessed.

The possible role of deception should also be considered in evaluating hunting behaviour in trophy hunting systems. Generally, apparently costly signals are potentially subject to cheating by modern humans [59]. In our system, with only minimal real risk of failure or injury, guided hunters might simply pay money to buy experiences that serve to deceive signal recipients. We suspect that signals broadcast by contemporary hunters are no longer honestly linked to cognitive or physical qualities due to expert guides and efficient weaponry [36,51]. Accordingly, all that is required for such deception to occur is for hunters to desire costly prey. Whereas in the past, underlying qualities were necessary to hunt costly prey, today's guided hunters can simply buy such opportunities in a context with no obvious fitness-related penalties of cheating. If true, this behaviour is similar to the purchase and display of luxury or brand-named goods and activities, termed 'conspicuous consumption' by sociologists [60].

Regardless of the underlying behavioural context, hunters showing increased desire to kill large carnivores might provide additional insight into why large carnivores have been [61–63] and continue to be [36] exploited at such high rates. There is disagreement on the impact of trophy hunting on population dynamics of prey [64–66]. Our work and that of others [15–19] suggest that management strategies for vulnerable wildlife should also consider how hunting policy might alter the potential costs, signals, and social benefits to hunters.

Data accessibility. All raw data and R code for analysis are available from the Dryad Digital Repository: https://doi.org/10.5061/dryad.vd34vr3 [67].

Authors' contributions. I.M. and C.T.D. conceived and designed the study. I.M. collected the data. I.M. and A.W.B. analysed the data and prepared figures. I.M., C.T.D. and A.W.B. wrote the manuscript. All authors gave final approval for publication.

Competing interests. We declare we have no competing interests.

Funding. I.M. was supported by a University of Victoria Jamie Cassels Undergraduate Research Award and a Natural Science and Engineering Research Council (NSERC) Undergraduate Student Research Award. A.W.B. was supported by an NSERC Banting Postdoctoral Fellowship. C.T.D. was supported by NSERC Discovery (grant no. 435683).

Acknowledgements. We thank B. Starzomski and T. Dawson for their important insight at the early stages of this project. We also thank members of the UVic Applied Conservation Science Lab for their support and input throughout this project.

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
