## [Reviewer comments · Royal Society Open Science]

Review History

RSOS-182113.R0 (Original submission)

Review form: Reviewer 1 (Christopher R. von Rueden)

Is the manuscript scientifically sound in its present form?

Yes

Are the interpretations and conclusions justified by the results?

No

Is the language acceptable?

Yes

Is it clear how to access all supporting data?

Yes

Do you have any ethical concerns with this paper?

No

Have you any concerns about statistical analyses in this paper?

No

Recommendation?

Reject

Comments to the Author(s)

The authors look at prices of trophy hunts in North America to test a signaling theory of hunters' motivations. While there is great potential benefit here to inform conservation policy, I have some issues with application of signaling theory to this dataset. First, there is no determination of what hunters are signaling by targeting game that is rare, dangerous, etc. Second, paying more to go on a hunt does not clearly signal any underlying quality on part of these hunters, such as physical condition or intelligence. It may be that those with more money can afford such hunts, but it's unlikely the hunts are primarily responsible for indicating affluence of the hunters to their peers. A reputation model and not a costly signaling model per se may be the better theoretical approach. These theoretical concerns don't aid the largely null results, which add to our inability to take much from the paper. I elaborate on these comments below.

Page 3, Line 28: the behavior you describe (paying large fees and traveling to unfamiliar areas) is clearly not the aspects of the behavior that evolved. Targeting large individuals and rare taxa may be (as you describe at the beginning of the paragraph) but your references are for modern trophy hunting not archaeological or ethnographic evidence from foragers. You should cite evidence here that such hunting behavior is characteristic of humans beyond modern trophy hunting. The Foraging Spectrum book by Kelly is a good overview. As well as studies you cite later, from Hawkes, Bliege Bird, etc.

Page 3, Line 52: Costly signaling means signalers must pay an extra cost (the handicap) at equilibrium, and high quality signalers can better afford this marginal cost. So its not that only certain individuals can pay the cost- its that certain individuals can better afford the marginal cost.

Page 4, Line 26: say what you mean by the theory not being universally accepted, per the article you cite. I think the gist is that signals need not be costly to be honest. Rather, the potential cost of cheating can keep signals honest, or some signals can be difficult if not impossible to fake (i.e. indices), or honest signalers may benefit more than fakers. Can these possibilities explain hunting behavior? Comment on this.

Page 5, Line 5: in traditional societies, there is typically a substantial fraction of hunted game that is kept by hunters and given to their families (see work by Michael Gurven). Thus hunting is unlikely to be purely or even principally a costly signal, or one that is "wasteful". Indeed, much of the higher reproductive success attributable to better hunters may flow through family provisioning. See Gurven and von Rueden "Hunting social status and biological fitness". Also see paper by Wood and Marlowe "Household and kin provisioning by Hadza men" that shows Hadza hunters target both small and large game and game of all sizes is widely shared. Hunting motivations are multi-faceted.

Page 5, Line 15: What is hunting actually signaling that begets higher reproductive success? There is no good evidence of this. The authors you cite only speculate, so be clear on this. An alternative to the costly signaling account of hunting is a reputation account. For details see "Costly signaling and the handicap principle in anthropology and zoology: a review" by Duncan Stibbard-Hawkes. It also makes points similar to comments above.

Page 5, Line 45: In the context of modern trophy hunting, affluence is unlikely to be signaled principally by killing particular prey. Taking the hunting trip itself is an indication of affluence, and those spending a lot on hunting are also likely to spend a lot on other material goods, e.g. billionaires who trophy hunt the African Big Five. Also, physical competency and cognitive ability are not necessarily required when guides and vehicles and sophisticated weaponry are on hand- as you recognize.

Page 6, Line 22: status among whom? Among Safari Club members, but normative disapproval of trophy hunting (e.g. Cecil the Lion) suggests status benefits are not likely to be widespread.

Page 6, Line 29: I'd suggest you set up an alternative prediction, that rarity associates with higher price because of supply/demand based on quotas to conserve the population. And this rarity need not be primarily a consequence of the Anthropogenic Allee Effect. But you wouldn't predict such quotas would track body size, danger, difficulty to hunt, or carnivory per se (independent of rarity).

Page 17, Line 17: Investigating the motives of trophy hunters is important for conservation efforts, perhaps particularly so in North America, but in many parts of the world subsistence hunting has bigger consequences for endangered wildlife. To what extent does trophy hunting provide benefits for conservation, e.g. revenue is funneled into wildlife education or protected area management or lobbying.

Review form: Reviewer 2

Is the manuscript scientifically sound in its present form?

Yes

Are the interpretations and conclusions justified by the results?

Yes

Is the language acceptable?

Yes

Is it clear how to access all supporting data?

Yes

Do you have any ethical concerns with this paper?

No

Have you any concerns about statistical analyses in this paper?

I do not feel qualified to assess the statistics

Recommendation?

Accept with minor revision (please list in comments)

Comments to the Author(s)

This is an interesting manuscript that makes a modest but solid contribution to the literatures on human signaling and commercial big game hunting.

My only problem with the manuscript is with the authors' presentation of costly signaling theory. They seem to equate costly signals with handicaps. That is not correct. One way to make a signal costly, and thus more likely to overcome a receiver's skepticism regarding the accuracy of the information it is trying to convey, is to make it a handicap. But that's not the only way to do it, and I don't think paying a lot of money to kill a bear or an elk qualifies as a handicap. I recommend the authors re-read the Grose article that they already cite. These two citations might also be useful:

Cronk, L., 2005. The application of animal signaling theory to human phenomena: some thoughts and clarifications. *Social Science Information*, 44(4), pp.603-620.

Smith, J.M. and Harper, D., 2003. *Animal signals*. Oxford University Press.

Regarding the Grose article, I think his problem is not so much with costly signaling theory in general, as implied by the way the authors of this manuscript cite it, but rather with how the handicap principle has been applied or misapplied.

Finally, I agree that the example of turtle hunting by the Meriam is a good one to use. The following article makes the same kind of point based on data from another island society, so the authors might want to cite it, as well:

Sosis, R., 2000. Costly signaling and torch fishing on Ifaluk Atoll. *Evolution and Human Behavior*, 21(4), pp.223-244.

Decision letter (RSOS-182113.R0)

20-Feb-2019

Dear Ms Mihalik:

Manuscript ID RSOS-182113 entitled "Big game hunters pay more to target larger-bodied carnivores: insight from costly signalling theory" which you submitted to Royal Society Open Science, has been reviewed. The comments from reviewers are included at the bottom of this letter.

In view of the criticisms of the reviewers, the manuscript has been rejected in its current form. However, a new manuscript may be submitted which takes into consideration these comments.

Please note that resubmitting your manuscript does not guarantee eventual acceptance, and that your resubmission will be subject to peer review before a decision is made.

Your resubmitted manuscript should be submitted by 20-Aug-2019. If you are unable to submit by this date please contact the Editorial Office.

on behalf of Dr Claudia Wascher (Associate Editor) and Kevin Padian (Subject Editor)
openscience@royalsociety.org

Associate Editor Comments to Author (Dr Claudia Wascher):

Associate Editor: 1

Comments to the Author:

The presented manuscript aims at testing whether predictions derived from costly signalling theory would apply to modern day big game hunting in humans and investigated whether the price to hunt a specific species would depend on the rarity, danger, or difficulty imposed by the species. The authors did find body size to positively affect hunting price. Both reviewers find the contribution potentially interesting, but raise concerns regarding the general framing of the manuscript within signalling theory. These concerns need to be addressed prior to potential publication of the manuscript.

Editor comments:

I agree that the results of this manuscript will be of great interest to conservationists (and I think will raise awareness among the general public). However, there are concerns about framing this within the general theory of costly signaling, and reviewers and editors think that this needs some rethinking. If you choose to resubmit, please address these concerns fully in your response. Thanks for submitting.

Reviewers' Comments to Author:

Reviewer: 1

Comments to the Author(s)

The authors look at prices of trophy hunts in North America to test a signaling theory of hunters' motivations. While there is great potential benefit here to inform conservation policy, I have some issues with application of signaling theory to this dataset. First, there is no determination of what hunters are signaling by targeting game that is rare, dangerous, etc. Second, paying more to go on a hunt does not clearly signal any underlying quality on part of these hunters, such as physical condition or intelligence. It may be that those with more money can afford such hunts, but it's unlikely the hunts are primarily responsible for indicating affluence of the hunters to their peers. A reputation model and not a costly signaling model per se may be the better theoretical approach. These theoretical concerns don't aid the largely null results, which add to our inability to take much from the paper. I elaborate on these comments below.

Page 3, Line 28: the behavior you describe (paying large fees and traveling to unfamiliar areas) is clearly not the aspects of the behavior that evolved. Targeting large individuals and rare taxa may be (as you describe at the beginning of the paragraph) but your references are for modern trophy hunting not archaeological or ethnographic evidence from foragers. You should cite evidence here that such hunting behavior is characteristic of humans beyond modern trophy hunting. The Foraging Spectrum book by Kelly is a good overview. As well as studies you cite later, from Hawkes, Bliege Bird, etc.

Page 3, Line 52: Costly signaling means signalers must pay an extra cost (the handicap) at equilibrium, and high quality signalers can better afford this marginal cost. So its not that only certain individuals can pay the cost- its that certain individuals can better afford the marginal cost.

Page 4, Line 26: say what you mean by the theory not being universally accepted, per the article you cite. I think the gist is that signals need not be costly to be honest. Rather, the potential cost of cheating can keep signals honest, or some signals can be difficult if not impossible to fake (i.e. indices), or honest signalers may benefit more than fakers. Can these possibilities explain hunting behavior? Comment on this.

Page 5, Line 5: in traditional societies, there is typically a substantial fraction of hunted game that is kept by hunters and given to their families (see work by Michael Gurven). Thus hunting is

unlikely to be purely or even principally a costly signal, or one that is “wasteful”. Indeed, much of the higher reproductive success attributable to better hunters may flow through family provisioning. See Gurven and von Rueden “Hunting social status and biological fitness”. Also see paper by Wood and Marlowe “Household and kin provisioning by Hadza men” that shows Hadza hunters target both small and large game and game of all sizes is widely shared. Hunting motivations are multi-faceted.

Page 5, Line 15: What is hunting actually signaling that begets higher reproductive success? There is no good evidence of this. The authors you cite only speculate, so be clear on this. An alternative to the costly signaling account of hunting is a reputation account. For details see “Costly signaling and the handicap principle in anthropology and zoology: a review” by Duncan Stibbard-Hawkes. It also makes points similar to comments above.

Page 5, Line 45: In the context of modern trophy hunting, affluence is unlikely to be signaled principally by killing particular prey. Taking the hunting trip itself is an indication of affluence, and those spending a lot on hunting are also likely to spend a lot on other material goods, e.g. billionaires who trophy hunt the African Big Five. Also, physical competency and cognitive ability are not necessarily required when guides and vehicles and sophisticated weaponry are on hand- as you recognize.

Page 6, Line 22: status among whom? Among Safari Club members, but normative disapproval of trophy hunting (e.g. Cecil the Lion) suggests status benefits are not likely to be widespread.

Page 6, Line 29: I'd suggest you set up an alternative prediction, that rarity associates with higher price because of supply/demand based on quotas to conserve the population. And this rarity need not be primarily a consequence of the Anthropogenic Allee Effect. But you wouldn't predict such quotas would track body size, danger, difficulty to hunt, or carnivory per se (independent of rarity).

Page 17, Line 17: Investigating the motives of trophy hunters is important for conservation efforts, perhaps particularly so in North America, but in many parts of the world subsistence hunting has bigger consequences for endangered wildlife. To what extent does trophy hunting provide benefits for conservation, e.g. revenue is funneled into wildlife education or protected area management or lobbying.

Reviewer: 2

Comments to the Author(s)

This is an interesting manuscript that makes a modest but solid contribution to the literatures on human signaling and commercial big game hunting.

My only problem with the manuscript is with the authors' presentation of costly signaling theory. They seem to equate costly signals with handicaps. That is not correct. One way to make a signal costly, and thus more likely to overcome a receiver's skepticism regarding the accuracy of the information it is trying to convey, is to make it a handicap. But that's not the only way to do it, and I don't think paying a lot of money to kill a bear or an elk qualifies as a handicap. I recommend the authors re-read the Grose article that they already cite. These two citations might also be useful:

Cronk, L., 2005. The application of animal signaling theory to human phenomena: some thoughts and clarifications. *Social Science Information*, 44(4), pp.603-620.

Smith, J.M. and Harper, D., 2003. *Animal signals*. Oxford University Press.

Regarding the Grose article, I think his problem is not so much with costly signaling theory in

general, as implied by the way the authors of this manuscript cite it, but rather with how the handicap principle has been applied or misapplied.

Finally, I agree that the example of turtle hunting by the Meriam is a good one to use. The following article makes the same kind of point based on data from another island society, so the authors might want to cite it, as well:

Sosis, R., 2000. Costly signaling and torch fishing on Ifaluk Atoll. *Evolution and Human Behavior*, 21(4), pp.223-244.

Author's Response to Decision Letter for (RSOS-182113.R0)

See Appendix A.

RSOS-191231.R0

Review form: Reviewer 1 (Christopher R. von Rueden)

Is the manuscript scientifically sound in its present form?

Yes

Are the interpretations and conclusions justified by the results?

Yes

Is the language acceptable?

Yes

Do you have any ethical concerns with this paper?

No

Have you any concerns about statistical analyses in this paper?

No

Recommendation?

Accept as is

Comments to the Author(s)

The changes addressed my concerns- looks good.

Review form: Reviewer 2

Is the manuscript scientifically sound in its present form?

Yes

Are the interpretations and conclusions justified by the results?

Yes

Is the language acceptable?

Yes

Do you have any ethical concerns with this paper?

No

Have you any concerns about statistical analyses in this paper?

No

Recommendation?

Accept as is

Comments to the Author(s)

The authors have taken the reviewers' comments to heart and greatly clarified their manuscript. The distinction between "cost" and "price" is especially helpful.

The only change I would recommend is the deletion of the subtitle. I does not seem necessary. The title alone summarizes the manuscript's main finding.

Decision letter (RSOS-191231.R0)

29-Jul-2019

Dear Ms Mihalik

On behalf of the Editor, I am pleased to inform you that your Manuscript RSOS-191231 entitled "Big game hunters pay more to target larger-bodied carnivores: money, guns, and costly signals in non-subsistence systems" has been accepted for publication in Royal Society Open Science subject to minor revision in accordance with the referee suggestions. Please find the referees' comments at the end of this email.

The reviewers and Subject Editor have recommended publication, but also suggest some minor revisions to your manuscript. Therefore, I invite you to respond to the comments and revise your manuscript.

- Ethics statement

- Data accessibility

If you wish to submit your supporting data or code to Dryad (<http://datadryad.org/>), or modify your current submission to dryad, please use the following link:
<http://datadryad.org/submit?journalID=RSOS&manu=RSOS-191231>

- Competing interests

- Authors' contributions

- Acknowledgements

- Funding statement

Because the schedule for publication is very tight, it is a condition of publication that you submit the revised version of your manuscript before 07-Aug-2019. Please note that the revision deadline will expire at 00.00am on this date. If you do not think you will be able to meet this date please let me know immediately.

on behalf of Dr Claudia Wascher (Associate Editor) and Kevin Padian (Subject Editor)
openscience@royalsociety.org

Associate Editor Comments to Author (Dr Claudia Wascher):

The authors have significantly revised the manuscript alongside with previous comments by the reviewers. The reviewers recommend the manuscript to be accepted for publication, they only ask the subtitle to be deleted.

Reviewer comments to Author:

Reviewer: 1

Comments to the Author(s)

The changes addressed my concerns- looks good.

Reviewer: 2

Comments to the Author(s)

The authors have taken the reviewers' comments to heart and greatly clarified their manuscript. The distinction between "cost" and "price" is especially helpful.

The only change I would recommend is the deletion of the subtitle. I does not seem necessary. The title alone summarizes the manuscript's main finding.

Author's Response to Decision Letter for (RSOS-191231.R0)

See Appendix B.

Decision letter (RSOS-191231.R1)

13-Aug-2019

Dear Ms Mihalik,

I am pleased to inform you that your manuscript entitled "Trophy hunters pay more to target larger-bodied carnivores" is now accepted for publication in Royal Society Open Science.

Royal Society Open Science operates under a continuous publication model (<http://bit.ly/cpFAQ>). Your article will be published straight into the next open issue and this will be the final version of the paper. As such, it can be cited immediately by other researchers.

As the issue version of your paper will be the only version to be published I would advise you to check your proofs thoroughly as changes cannot be made once the paper is published.

Kind regards,

on behalf of Dr Claudia Wascher (Associate Editor) and Kevin Padian (Subject Editor)
openscience@royalsociety.org

Appendix A

1 **Big game hunters pay more to target larger-bodied**
**carnivores: insight from costly signalling money,**
**guns, and costly signalling signals theory in non-**
**subsistence systems**

**Iлона Mihalik^{1,2,*}, Andrew W. Bateman^{1,2}, and Chris T. Darimont^{1,2}**

¹Department of Geography, University of Victoria, Victoria, British Columbia, Canada

²Raincoast Conservation Foundation, Sidney, British Columbia, Canada

*corresponding author: ilonammillie@gmail.com

Formatted: Section start: Continuous, Numbering:
Continuous

**Keywords:** body size, costly signalling, trophy hunting, wildlife harvest, exploitation,
carnivore, ~~size selective harvesting~~

1. Abstract

~~Hunters~~Human hunters often target species that require resource investment disproportionate to
associated nutritional rewards. ~~CA~~Although a controversial theory, ~~ostly~~Costly signalling theory
provides a potential explanation, proposing that ~~hunters~~ theory may provide an explanation if
~~humans~~suggests that ~~subsistence and non-subsistence~~ hunters evolved behaviour to target species
that impose high ~~ecological~~ costs (e.g. higher failure and injury risks, lower consumptive returns)
because it ~~signals~~, thereby ~~thus~~ ~~signaling to potential mates or competitors~~ ~~an~~their ability to
absorb ~~such~~ costly behaviour. If costly signalling is relevant to contemporary 'big game'
hunters, ~~and if prices for guided hunts reflect desirability~~, we would expect ~~non-subsistence~~
hunters to pay ~~more~~ higher prices to hunt taxa with higher ~~associated-perceived~~ ~~ecological~~ costs,
~~(e.g. higher risks of failure risk, risk of and injury)~~. Accordingly, ~~among guided hunters we~~
hypothesized that ~~hunt~~ prices would be higher for taxa ~~with higher perceived ecological costs:~~
~~those that are that are~~ larger-bodied, rarer, carnivorous, or described as dangerous or difficult to
hunt. In a data set on 721 guided hunts for fifteen North American ~~big game species~~large
~~mammals~~, prices listed on ~~the internet line did not vary with rarity or perceived danger or~~
~~difficulty but~~ increased with body size ~~in carnivores~~ (from approximately \$550 USD/day to
\$1800 USD/day across the observed range) ~~of body mass~~ ~~in carnivores~~. This pattern suggests
that elements of costly signals ~~may~~ing persists among ~~contemporary noncontemporary non-~~
~~subsistence~~ hunters. Persistence might simply relate to deception, given that ~~despite~~ ~~d~~Despite
~~unlikely~~ signal honesty and ~~any~~ fitness benefits ~~being~~ are unlikely in such a ~~radically~~ in trophy
~~hunting systems~~radically different ~~social, and ecological and technological~~ conditions compared

Formatted: Font:

~~with ancestral environments in which hunting behaviour evolved... More broadly over. We did~~
~~not find a signal of rarity or perceived danger or difficulty. If~~ larger-bodied carnivores are
generally more desirable to hunters ~~due to perceived benefits of costly signalling~~, then
conservation and management strategies should consider not only the ecology of the hunted but
also the motivations of hunters.

2. Introduction

The behaviour of ~~human human~~ hunters and fishers diverges substantially from other predators
of vertebrate prey. Instead of targeting mainly juvenile or otherwise vulnerable individuals,
humans ~~(and more often, males)~~, typically seek large ~~taxa, as well as large, and~~ reproductive-
aged individuals ~~within populations~~¹⁻⁵, ~~phenotypes targets a behaviour also found in phenotypes~~
~~also targeted-sought by early human groups~~⁶. This distinct pattern of hunting behaviour is likely
~~shaped by a suite of multiple of adaptive selective forces~~⁷; for example, ~~i.~~ In subsistence
~~societies, provisioning-sharing~~ targeting large prey items may be motivated by kin provisioning⁸⁻
¹¹ ~~among kin family members is clearly important (has likely been favoured by kin selection;~~
~~“provisioning model”;~~ whereas, widely ~~Likewise, sOn the other hand, sSharing~~⁹). Likewise,

Commented [D1]: Best place to plant the other two theories, so we can move on...this works to explain why LARGE items are targeted

Formatted: Superscript

Formatted: Highlight

~~sharing meat from large items prey~~ beyond kin, and expecting the same in return. ~~may likely~~
~~presents an example of follow~~ (i.e. 'reciprocal altruism'^{12,13}),
~~Additional patterns of hunting behaviour~~ have informed other evolutionary models
~~te~~explanations underlying ~~explain~~ hunting behaviour. Within traditional hunter-gatherer groups,
~~for example, male hunters often target species with a highly variable caloric payoff over more~~
~~reliably or safely acquired alternatives~~¹⁴⁸. ~~Similarly, pParticularly-MoreoverSimilarly,~~
~~particularly in trophy-hunting contexts, contemporary huntershunters~~ Additionally, humans
often ~~similarly~~ pursue taxa that are rare¹⁵⁻¹⁹²⁻⁶ ~~and have historically exploited~~ populations
(especially terrestrial carnivores) at high rates compared to non-human predators²²⁷. ~~To seek~~
~~their prey, modern hunters often are often motivated to~~ travel to unfamiliar areas and pay large
~~fees to travel to unfamiliar areas and hire specialist guides~~⁸. Prices for guided hunts can be
substantial, ranging from several hundred to many thousands of US dollars (USD) per day²⁴. ~~In~~
~~addition, permits to hunt some species can reach tens of thousands of USD~~⁹. ~~Additionally,~~
~~owingOowing~~ to restrictions on meat exports, and to the targeting of seldom-eaten species, such
as large carnivores, ~~professionally modern~~ the large investments required for guided hunters
~~frequently seekare oftenseek prey, made~~ without the intention of receiving nutrition, the primary
benefit of predation in the wild. Such ~~seemingly~~
~~counter intuitive~~ ~~seemingly inefficient~~ behaviour begs the question: how did ~~such such~~
~~seemingly inefficient hunting~~ behaviour evolve, and why might it ~~persistecontinue continuepersist~~
today?

Formatted: Not Highlight

Formatted: Font: Not Italic

Commented [D2]: Why is this changed to past tense?

Ostensibly wasteful investments by animals have long intrigued researchers, inspiring theory,
empirical investigation, ~~theory~~, and debate. Darwin²⁴⁰, for example, questioned what drove the
evolution of extravagant traits in males, such as the large tails of peacocks (*Pavo* spp.) and
antlers of deer (Cervidae). Zahavi²⁴¹ proposed that time-consuming, risky, inefficient, or
otherwise ‘handicapping’ traits or activities could be interpreted as ‘costly signals.’ Costly
signalling theory suggests that a costly signal (~~or ‘handicap’~~) reflects the capacity of the signaller
to bear the cost, therefore providing honest information to potential mates and competitors about
the underlying quality of the signaller²⁴¹ (~~e.g., (e.g. the ‘strategic cost’²²).~~ The theory suggests
that honesty ~~Honesty~~ is maintained through the differential costs and benefits of signal
production; individuals of higher quality are thought to ~~be better~~ affordable to pay ~~afford~~ the
larger costs associated with more ~~elaborate attractive~~ signals; for lower-quality individuals,
~~while~~ the costs outweigh the benefits and signals are difficult to fake ~~for lower-quality~~
~~individuals¹² individuals^{12,132-24}~~. Under this framework, evolutionary benefits flow to ~~higher-~~
quality ~~the signallers (as well as and the signal recipients), the signaller. Individuals that can afford~~
~~costly signals attract mates or and accrue social status, which can increase access to resources~~
~~(e.g., foods, material goods, approval from peers, knowledge)^{11,14}~~. For example, in avian
courtship displays male birds subject themselves to predation risk by singing or dancing in the
open during sexual displays, signalling that they have underlying qualities that permit them to
absorb the energetic and predation-risk costs of the display²⁴¹. ~~Similarly, in~~ In human systems,
costly signalling has been used to explain behaviour associated with artistic elaboration,
ceremonial feasting, body modification, and monumental architecture^{5,2515}. Individuals that can
afford costly signals can attract mates or accrue social status, which can increase access to

~~resources (e.g., foods, material goods, approval from peers, knowledge)^{21,26}. Using similar logic,~~
~~sociologists use the term ‘conspicuous consumption’ to describe the purchase and display of~~
~~luxury goods and activities¹⁶. Although costly signalling and associated theory is not universally~~
~~accepted¹⁷, it provides a useful framework with which to evaluate apparently inefficient~~
~~behaviour.~~

~~Whereas data are limited and debate common, costly signalling has also been invoked~~
~~to explain hunting behaviour in some human subsistence-hunting systems, although relevant~~
~~data are limited and debate is common^{10, 27-29}. According to the theory in this context, when~~
~~subsistence hunters target Hunting behaviour by humans is multifaceted and not mutually-~~
~~exclusive, often motivated by provisioning oneself, kin, and/or through reciprocal altruism, while~~
~~certain behaviour Hunting by humans has also been considered in a costly signalling~~
~~framework. Even in the case of subsistence hunting within traditional hunter-gather groups, male~~
~~hunters often target species with a highly variable caloric payoff over more reliably or safely~~
~~acquired alternatives¹⁸. In targeting items with high ecological costs, they (i.e. risk of~~
~~failure, injury, etc), such hunters they would honestly signal their ability to absorb the~~
~~costs^{149,3018}. Thus, according to this theory, hunting itself becomes-serves as the signal, and s,~~
~~uccessfully and successfully hunting-hunting-or hunting a species with higher costs, signals-~~
~~produces a higher quality-signal (akin to a more showy avian courtship display). Hunting of~~
marine turtles (*Chelonia mydas*) by the Meriam peoples of Murray Island, Northern Australia,
provides an example. There, diverse members of Meriam society collect marine turtles as they
crawl on the beach where they are easily captured; however, only reproductive-aged men

Commented [MOU3]: say what you mean by the theory not being universally accepted, per the article you cite. I think the gist is that signals need not be costly to be honest. Rather, the potential cost of cheating can keep signals honest, or some signals can be difficult if not impossible to fake (i.e. indices), or honest signalers may benefit more than fakers. Can these possibilities explain hunting behavior? Comment on this.

participate in off-shore turtle hunting, a costly activity (i.e., high risk of failure; increased risk of
injury; lower consumptive returns; high energetic, monetary, time investment costs)^{32,10,3224,245}.
When successful, these hunters rarely consume the meat themselves, and instead
provision community members at large feasts, arguably providing the public forum to signal the
hunters' underlying qualities that allow them to engage in such costly
behaviour³¹~~behaviour~~^{20,3224,245}. Successful Meriam turtle hunters earn social status and higher
reproductive success, providing rare evidence for fitness benefits associated with apparent costly
signalling in humans³¹~~humans~~^{20,3224}. Men from other hunter-gatherer societies suggested to
exhibit similar signalling behaviour, not easily explained by provisioning or reciprocal altruism
alone, include the Ache men of Eastern Paraguay³⁰, the Hadza men of Tanzania³³, and male torch
fishers of Ifaluk atoll³⁴. However, some criticisms~~criticisms~~Criticisms of these interpretations
include whether men's hunting patterns are truly suboptimal in terms of nutrient acquisition (e.g.
argued in the case of the Hadza men²⁷), and that Hadza²⁸ and Ache²⁹ men value provisioning
over showing-off their hunting ability, regardless of having dependent offspring. Additionally,
Others argue that fitness benefits gained by hunters are influenced by multiple pathways, rather
than just through showing off¹⁰.
Although a controversial theory when applied to~~Examining~~ Costly signalling may help to human
explain elements of~~subsistence~~non-subsistence -hunting, Examining seemingly wasteful
hunting behaviour among non-subsistence-hunters (hunting without the in the goal of providing
food, e.g. trophy hunting) offers~~developed world~~offers new opportunity to confront elements of
this controversial theory~~costly signalling~~. In particular, theory~~Given such controversy, whether~~
Costly signalling may help to explain or not seemingly wasteful inefficient non-subsistence

~~hunting behaviour that among non-subsistence hunters appear to conform to predictions of~~
~~costly signalling theory might provide fresh new insight into the discussion debate. In particular,~~
~~nNAs explained above, Nn, similar to behaviour observed in traditional hunter-gatherer groupsd.~~
~~Specifically, non-subsistence trophy hunters hunters would supposedly seemingly appear to~~
~~would incur substantial costs -- (in terms of high failure risk or risk of injury as well as low to nil~~
~~consumptive returns --) when they target large-bodied, carnivorous, rare, and/or dangerous or~~
~~difficult-to-hunt species. Specifically, weWe would expect increased failure risk via lower~~
~~hunter-encounter rates with larger and higher trophic-level animals, which tend to occur at lower~~
~~densities than small, low-trophic-level species³⁵²². Similarly, hunters likely encounter other rare~~
~~species of high conservation concern (with populations reduced by human impacts) less~~
~~frequently than abundant species of low conservation concern. In addition, species that are~~
~~dangerous or difficult to hunt are likely to increase failure and injury risk, posing another cost.~~
~~owing to injury or unsuccessful attempts. Moreover, hIn additionMoreover, human hunters often~~
~~kill seldom-eaten species (citation), species, such as carnivores, which includes the opportunity~~
~~cost of forgoing greater nutrition from hunting edible prey. Collectively, hunting inefficiently by~~
~~targetingTargeting such prey inefficient prey characteristics could signal a perceivedaperceivedn~~
~~ability to accept the costsobstaclecosts of higher failure and injury risk, as well as opportunity~~
~~costs, compared with targeting species that are more easily securedlocatsecured and offering~~
~~higher nutritional return. Throughout this paper, we use the term “cost” to refer to these~~
~~Though the signals shown off by these opportunity costs (lower nutritional returns) andas well~~
~~as these failure and hunters may no longer be honestly linked to cognitive/physical qualities (due~~
~~to expert guides and injury risks; in contrast costs efficient weaponry^{8,36}), and we use the term~~
~~“price” (see below) when referring to currencythe money hunters pay for perhaps not adaptive.~~

Formatted: Font: Bold

we argue the behaviour still persists due to benefits provided by costly signaling in previous
conditions (i.e. mismatch theory). Indeed, showing off risky game was perhaps once
advantageous throughout ancestral hunter-gatherer periods of human evolution (and still adaptive
in some modern-day hunter-gatherer groups^{20,21}), however, fitness benefits from hunting big
game today are difficult to measure within the developed world, even if the tendencies are still
ingrained. Although, elaborate a Similar to the avian courtship displays and offshore turtle
hunting in Meriam society, hunting behaviour that targets such costly prey might signal
otherwise hidden qualities, physical competency, or cognitive ability⁷. Awards from, and status
hierarchies within, from organizations with large followings like Safari Club International (SCI)
provide evidence of modern-day status benefits to signallers. SCI offers dozens of elaborate
awards that create status hierarchies among members; for guided hunts, example, to achieve the
World Hunting Award one must have already achieved 11 Grand Slam Awards, 17 diamond-
level Inner Circle Awards, and both the Fourth Pinnacle of Achievement and Crowning
Achievement Award²³.

Although the targeting of some big-game (i.e., relatively large species mammals hunted for sport)
by modern, contemporary, non-subsistence hunters appears to include many elements of costly
signalling behaviour, there have been no empirical evaluations of the theory in this context
context's relevance. If such trophy hunting behaviour is an example of reflects elements of
costly signalling persistent that persists among in modern among contemporary human hunters, we
would predict that species with high perceived costs should be more desirable to hunters because
they could signal a greater ability to absorb the accept costs. Accordingly, assuming that
market demand (which could also be influenced by lower supply in the case of rare species)

Commented [MOU4]: In the context of modern trophy hunting, affluence is unlikely to be signaled principally by killing particular prey. Taking the hunting trip itself is an indication of affluence, and those spending a lot on hunting are also likely to spend a lot on other material goods, e.g. billionaires who trophy hunt the African Big Five. Also, physical competency and cognitive ability are not necessarily required when guides and vehicles and sophisticated weaponry are on hand- as you recognize.

~~causes-influences~~ price to reflect desirability ~~--~~ (a common assumption¹⁵⁻²¹⁹⁻⁶ ~~==>~~ ~~--~~) we
hypothesized that hunt prices would be higher, ~~and accordingly hunters would be willing to pay,~~
for taxa with higher perceived ~~ecological~~ costs ~~of hunting.~~ We note that lower supply, through
~~rarity or hunting~~ restrictions, could also drive up prices, but we would not expect to find an
~~association with prey body size, or hunt danger, or difficulty in this case.~~ We confronted ~~this our~~
~~hypothesis using data from~~ guided big game trophy ~~hunter~~ hunter systems, where ~~These~~
~~hunters hire specialist guides~~³⁶⁸. Prices for guided hunts can be substantial, ranging from several
~~hundred to many thousands of US dollars (USD) per day~~¹⁵³⁻¹⁷⁴. ~~In addition,~~ ~~permits to hunt some~~
~~species can reach tens of thousands of USD~~⁹. Specifically, using price charged per day for
guided hunts as an index, we predicted that species that are 1) large-bodied, 2) rare, 3)
carnivorous, and 4) described by Safari Club International (SCI)³⁷ as dangerous or difficult-to-
hunt, ~~would be priced higher.~~ ~~In other words, we predicted~~ ~~predicted that guided hunters would~~
~~pay higher prices for hunts that signal ecologically costlier behavior.~~

Commented [AL5]: Tangential and will cause problems

Formatted: Superscript

Formatted: Font: (Default) Times New Roman, Not Bold

Formatted: Font: (Default) Times New Roman, Not Bold

[revised manuscript text omitted]
 higher ~~perceived risk of failure risk-and injury, as well as and~~ low consumptive
returns, ~~and are therefore perceived as more costly, as we argue below~~.

Some patterns we observed differed from previously published findings. For one, the
jurisdiction-level ~~conservation status~~ (~~state or provincial-level within North America~~) of a
species (~~is our proxy for~~ rarity) did not affect price in our analysis. In contrast, larger-scale
conservation rankings (such as IUCN and/or CITES) have previously been found to correlate
with hunting price in Caprinae species¹⁵², Bovidae taxa^{163,174}, ungulates¹⁸⁵, and African felids¹⁹⁶.
Two explanations for why we did not detect a relationship might be relevant. First, the
jurisdiction-level rankings showed little variation across species ~~in our data set~~; only 18% (n =
27) of the species-jurisdiction combinations ranged from ‘Vulnerable’/‘Apparently Secure’
(~~S3/S4S3S4~~) to ‘Critically Imperiled’ (S1). ~~The remainder were not classified as at risk~~. Second,
hunters (especially when they commonly originate from afar) might not know or consider the
jurisdiction-level conservation status (~~that we which were~~ included in ~~our~~ models) of their targets,
whereas a species’ IUCN ranking is typically well-known and often included in their SCI record
book descriptions. However, only two species used in our study were ranked by the IUCN as
“Vulnerable”, while the remaining thirteen ranked as “Least Concern”.

We found that the presence of a ‘difficult and/or dangerous’ hunt description by
SCI³⁷ description²⁶ likewise had no statistical influence, ~~statistically~~, on price. This result
departed from our predictions, given that difficult and dangerous descriptions should increase the
perception of failure risk and risk of injury. We speculate that, unlike subsistence hunts (which
likely carry realistic and meaningful risk of failure), guided big game hunters in reality risk
relatively little in terms of failure owing to difficulty or danger. Contemporary hunters now
employ efficient killing technology to hunt prey at a safe distance^{8,36,51}. Indeed, while we
expected the perception of difficulty and danger to matter in terms of desirability, guided hunts
that pose real risks to safety might be relatively uncommon, and ~~customers guided clients (and~~
~~those to whom hunting signals are broadcast)~~ are likely to be aware of ~~know~~ this.

Our work has several potential limitations. Among them, we assume that prices charged to hunt
different species reflect desirability for hunters, an assumption commonly made in ~~the related~~
~~literature on the Anthropogenic Allee Effect (AAE)¹⁵²⁻¹⁹⁶. Similarly, we assumed rarity to be~~
~~more desirable owing to lower supply.~~ Additional factors are likely also involved. While we did
not address it in our study, due to the coarse state- or province-scale resolution of available data,
the cost of living (food, accommodation, and guiding) may also influence prices. Given that the
two largest carnivores (polar and grizzly bears) in our data-set occur at northern latitudes,
associated with remoteness and high costs of living, this was of concern. Accordingly, we ~~post~~
~~hoc~~ examined post hoc whether latitude could explain the high hunt prices observed for large
carnivores. While large carnivores do tend to occur at higher latitudes (Figure S4), we found no
statistical evidence that latitude drove hunt price for carnivores (Figure S5). Additionally, some
might argue that pursuing larger-bodied carnivores might have additional costs related to

searching for targets, given their naturally low density. This is possible, but we standardized our
price metric to daily rates, dealing with the possibility that lower density species might take
longer to locate. Furthermore, the use of an imputed mean for hunts without a listed duration,
calculated by using the mean hunt-length for a species-jurisdiction (combination of each species
in every North American province and state in which they occur), could lead to biased results
for carnivores (if they do indeed require additional search times). Finally, we acknowledge
Google's search results may vary across users and limit reproducibility⁵².

We ~~speculate-argue~~ that the relationship between body mass and price is evident only in
carnivores (Figure 1) because larger size in ~~these taxa~~ carnivores strongly signal increased
perceived measures of danger or, ~~rarity and other costs not accounted for in our analysis.~~
Specifically, although not captured in SCI descriptions, larger-bodied carnivores could give the
perception of increased danger; displaying a carcass of a predator could signal the absorbed costs
of interacting with animals that, compared to ungulates, are perceived as more dangerous if they
are larger-bodied. Additionally, larger-bodied carnivores are naturally rarer, owing to their
higher trophic position³⁵²². This dimension of rarity (*perceived rarity*⁵³³⁷) could be recognized
by hunters and could therefore serve as a better proxy for rarity ~~compared with~~ than conservation
status, especially on a continent where few hunted taxa are of ~~dire~~ conservation concern. Finally,
unlike herbivores, carnivores are generally not consumed, imposing the additional cost of
receiving no nutritional gains from kills. ~~Only~~ only the smaller bodied black bear (classified here
as a carnivore) is commonly eaten. Whereas these explanations are speculative, they generally
align with previous research that has found North American hunters display evidence of
'achievement satisfaction' (congruence of goals and outcomes regarding performance) more

commonly when sharing information about carnivore hunts compared to herbivore hunts. For
example, men posing with ~~ungulates bearing larger antlers~~~~ornaments~~~~antlers, and~~ carnivores of
any size, in hunting photographs have ~~greater~~ higher odds of displaying a ‘true smile’, an honest
signal of pleasure, compared to pictures with herbivore prey⁵⁴. pleasure³⁸. Additionally, in~~in~~
online discussion forums about hunting, men express achievement-oriented phrases more
frequently when describing carnivore hunts compared to ungulate hunts^{55,39}.

Our results, showing increased value placed by hunters on large-bodied prey, share similarities
with work conducted in other areas that adopted a different line of conceptual inquiry.
Specifically, the ~~AAE~~ anthropogenic Allee ~~Anthropogenic Allee effect (Effect (AAE) AAE)~~
~~describes emerges when~~ describes a phenomenon in which rare species become more desirable to
hunters¹⁵². In this context, others have similarly found that body size positively correlates with
hunting prices, specifically in ungulates¹⁸⁵ and African species¹⁶³. Our results thus increase the
scope of taxa and contexts involved in the pattern, suggesting, that, although not universal, the
desire of hunters~~hunters’ desire~~ to kill larger species exists across different environments,
cultures, conservation contexts, and communities of species available for hunting. This
observation of similar patterns across diverse systems of contemporary hunting suggests the
potential for an underlying evolutionary origin of the behaviours involved.

Might costly signalling explain the evolution and ~~maintenance~~ persistence of otherwise
inefficient behaviour in ~~and displays deep rooted~~ Costly of some contemporary hunters? The
theory is not universally accepted, argued by some to be particularly misapplied in studies of

~~contemporary human behaviour¹⁷. One important consideration is that apparently costly~~
~~signalling/signalling human signals systems are potentially subject to cheating by modern humans~~
~~One possibility in our systems is that guided hunters in fact pay money to buy experiences that in~~
~~effect deceive recipients of costly hunting signals; despite a near absence of real risk of failure~~
~~and injury, they pay more to signal greater perceived costs. We/Additionally, wWe suspect that~~
~~signals broadcast by contemporary hunters are likely no longer honestly linked to~~
~~cognitive/physical qualities (due to expert guides and efficient weaponry^{8,36}).~~
~~Whereas/Whereas and associated theory, provides a useful framework with which to evaluate the~~
~~evolution and maintenance/persistence of apparently inefficient behaviour in trophy hunting~~
~~systems, but- care in use and interpretation is required. The theory is argued by some to have~~
~~been misapplied in contemporary human behavior studies⁵⁶. Given that our work only relates to~~
~~one prediction within the costly signalling framework (that hunters should be willing to pay more~~
~~to hunt species perceived as imposing higher ecological costs), further work is required to~~
~~elucidate the potential relevance of costly signalling in this context/the theory in this context. Any~~
~~We did not. -Although not evaluate any fitness benefits of costly considered in our study as a~~
~~'cost', prices paid to hunt a species represent significant costs in today's monetary-based~~
~~currency. Thus, guided hunters might additionally signal their financial ability to afford such~~
~~expensive pursuits^{16,7}. A fitness benefits of costly signalling among to guided hunters, for~~
~~example, has not been evaluated but also such benefits seem unlikely. Persistence of~~
~~evolutionarily-mismatched behaviours, however, is common in contemporary behaviour/human~~
~~society (e.g. gambling⁵⁷, and risk-taking in adolescents⁵⁸) and perhaps seems likely/likely in our~~
~~this case, given such rapid and extraordinary change in likely differences between current social~~

Formatted: Font: Not Italic

Formatted: Font: Not Italic

~~and ecological conditions environments compared to and the~~ ancestral environments in which
hunting behaviour evolved. ~~Though the we suspect that signals shown off broadcast by~~
~~these contemporary hunters may are likely no longer be honestly linked to cognitive/physical~~
~~qualities (due to expert guides and efficient weaponry^{8,36} purchased by hunters), and perhaps~~
~~not adaptive, we argue the behaviour still Persistence of evolutionarily mismatched behaviours;~~
~~however, is common (FeF) and perhaps likely in our case, given persists such rapid and~~
~~extraordinary change in social and ecological conditions compared to ancestral environments in~~
~~which hunting behaviour evolved.~~ However, ~~due to benefits provided by costly signaling in~~
~~previous conditions (i.e. mismatch theory). Indeed, showing off risky game was perhaps once~~
~~advantageous throughout ancestral hunter-gatherer periods of human evolution (and still adaptive~~
~~in some modern-day hunter-gatherer groups^{20,21}), however, fitness benefits from hunting big~~
~~game today are difficult to measure within the developed world, even if the tendencies are still~~
~~ingrained. Although~~ Additionally, any fitness benefits of costly signaling among guided hunters
~~also seem unlikely. Instead~~ However, ~~elaborate awards from, and status hierarchies within,~~
organizations with large followings (e.g. like Safari Club International (SCI)) provide evidence of
modern-day ~~status-social~~ benefits to signallers. Although there is general societal disapproval for
trophy hunting. SCI offers dozens of ~~elaborate~~ awards that create status hierarchies among
members; for example, to achieve the *World Hunting Award* one must have already achieved 11
*Grand Slam Awards*, 17 diamond-level *Inner Circle Awards*, and both the *Fourth Pinnacle of*
*Achievement* and *Crowning Achievement Award*³⁸. ~~Beyond these SCI award recipients, however,~~
~~normative general societal disapproval might in fact function to shame trophy hunters⁷.~~ ~~Future,~~
~~future~~ studies could assess the relationships between costs absorbed and measures of related
social status earned; ~~withh.~~ With an online and increasingly globalized audience, examinations of

Formatted: Font: Not Italic

Formatted: Font: Not Bold

Formatted: Highlight

Formatted: Font: Italic

the support (e.g., 'likes' or other positive feedback received on social media platforms) in
~~different big game~~ hunting contexts could yield new insight. Work is also required to examine
the potential benefits flowing to ~~audience members (the signal recipients)~~, asking what
information on signaller quality might be assessed.

~~Additional possibilities need to~~The possible role of deception should also be considered in
~~evaluating hunting behaviour in the~~trophy hunting systems. ~~Specifically, costly signaling~~
~~theory is argued by some to be particularly misapplied in studies of contemporary human~~
~~behaviour~~¹⁷. ~~One important consideration is that~~Generally, apparently costly signals are
~~potentially subject to cheating by modern humans~~^{59(Cronk, 2005)}. ~~In our system, with only minimal~~
~~real risk of failure or injury, guided hunters might simply pay money to buy experiences that in~~
~~effect serve to deceive signal recipients, of costly hunting signals; despite a near absence of real~~
~~risk of failure and injury, hunters pay to signal costs.~~We suspect that signals broadcast by
~~contemporary hunters are likely no longer honestly linked to cognitive or physical qualities (due~~
~~to expert guides and efficient weaponry~~^{368,5136}). Accordingly, all that is required for such
~~deception to occur is for hunters to desire costly prey. Whereas in the past, underlying qualities~~
~~were necessary to hunt costly prey, today's guided hunters can simply buy such opportunities via~~
~~a mechanism (money) and context in a context with no fitness-related penalties of cheating. If~~
~~true, this behaviour is similar to the purchase and display of luxury or brand-named goods and~~
~~activities, termed 'conspicuous consumption' by sociologists~~⁶⁰.

Regardless of underlying behavioural context, hunters showing increased desire to kill large
carnivores might provide additional insight into why large carnivores have ~~been~~ ~~been exploited~~
~~at such high rates in the past~~⁴⁰⁻⁶¹⁻⁶³~~42~~ and ~~present~~³⁶ ~~continue to be~~³⁶ exploited at such high rates.-
~~Although there is~~ There is disagreement among on the impact of trophy hunting on population
~~dynamics of prey~~⁶⁴⁻⁶⁶. ~~O~~ ~~The patterns revealed in our~~ work and that of others¹⁵²⁻¹⁹⁶ suggest that
management strategies for vulnerable wildlife should also consider how ~~not only population~~
~~dynamics of prey but also how different~~ hunting policy might alter the potential costs, signals,
and social benefits to hunters.

~~Using similar logic, sociologists use the term ‘conspicuous consumption’ to describe the~~
~~purchase and display of luxury goods and activities~~¹⁶.

**Data Accessibility**

All raw data and R code for analysis are included in the Dryad Depository:

<https://datadryad.org/review?doi=doi:10.5061/dryad.b1k2979>

**Research Ethics – *Heading does not apply to our manuscript***

Research Ethics was not applicable. We were not required to complete an ethical assessment

prior to conducting the research.

**Animal Ethics – *Heading does not apply to our manuscript***

Animal Ethics was not applicable. We were not required to complete an ethical assessment prior

to conducting the research.

**Permission to Carry out Fieldwork – *Heading does not apply***

***to our manuscript***

No permissions to carry out fieldwork were required prior to conducting the research.

**Funding Statement**

IM was supported by a University of Victoria Jamie Cassels Undergraduate Research Award and

a Natural Science and Engineering Research Council (NSERC) Undergraduate Student Research

Award. AWB was supported by a NSERC Banting Postdoctoral Fellowship. CTD was supported
by NSERC Discovery Grant 435683.

**Competing Interests**

The authors declare no competing interests.

**Author Contributions**

IM and CTD conceived and designed the study. IM collected the data. IM and AWB analyzed
the data and prepared figures. IM, CTD, and AWB wrote the manuscript.

[revised manuscript text omitted]
Jacobs JW, James J, Volk AA, Wilson DS. 2012. The evolutionary basis of risky adolescent
behavior: implications for science, policy, and practice. *Dev. Psychol.* **48**, 598. (doi:
10.1037/a0026220)
- 59. Cronk L. 2005. The application of animal signaling theory to human phenomena: some
thoughts and clarifications. *Soc. Sci. Inf.* **44**, 603-620. (doi: 10.1177/0539018405058203)

Formatted: Indent: Left: 1.5 pi, No bullets or numbering

Formatted: Font: Font color: Text 1, Expanded by 0.1 pt

60. Veblen T. 1899. *Theory of the leisure class: an economic study in the evolution of*
institutions. New York, NY:Macmillan.
61. Estes JA et al. 2011. Trophic Downgrading of Planet Earth. *Science* **333**, 301-306.
(doi:10.1126/science.1205106)
62. Barnosky AD, Koch PL, Feranec RS, Wing SL, Shabel AB. 2004. Assessing the Causes of
Late Pleistocene Extinctions on the Continents. *Science* **306**, 70-75.
(doi:10.1126/science.1101476)
63. Ray J, Redford KH, Steneck R, Berger J. (eds.) 2005. *Large carnivores and the conservation*
of biodiversity. Washington DC:Island Press.
64. Batavia C, Nelson MP, Darimont CT, Paquet PC, Ripple WJ, Wallach AD. 2019. The
elephant (head) in the room: A critical look at trophy hunting. *Cons. Lett.* **12**, e12565. (doi:
doi.org/10.1111/conl.12565)
65. Di Minin E, Leader-Williams N, Bradshaw CJ. 2016. Banning trophy hunting will exacerbate
biodiversity loss. *Trends Ecol. Evol.* **31**, 99-102. (doi: 10.1016/j.tree.2015.12.006)
66. Ripple WJ, Newsome TM, Kerley GI. 2016. Does trophy hunting support biodiversity? A
response to Di Minin et al. *Trends Ecol. Evol.* **31**, 495-496. (doi: 10.1016/j.tree.2016.03.011)

**References**

1. Stenseth NC, Dunlop ES. 2009. Evolution: Unnatural selection. *Nature* **457**, 803–804.
(doi:10.1038/457803a)
2. Courchamp F, Angulo E, Rivalan P, Hall RJ, Signoret L, Bull L, Meinard Y. 2006. Rarity
Value and Species Extinction: The Anthropogenic Allee Effect. *PLoS Biol.* **4**, e115.
(doi:10.1371/journal.pbio.0040415)
3. Johnson PJ, Kansky R, Loveridge AJ, Macdonald DW. 2010. Size, Rarity and Charisma:
Valuing African Wildlife Trophies. *PloS ONE* **5**, e12866.
(doi:10.1371/journal.pone.0012866)
4. Prescott GW, Johnson PJ, Loveridge AJ, Macdonald DW. 2012. Does change in IUCN status
affect demand for African bovid trophies? *Anim. Conserv.* **15**, 248–252. (doi:10.1111/j.1469-
1795.2011.00506.x)
5. Palazy L, Bonenfant C, Gaillard JM, Courchamp F. 2011a. Rarity, trophy hunting and
ungulates. *Anim. Conserv.* **15**, 4–11. (doi:10.1111/j.1469-1795.2011.00476.x)
6. Palazy L, Bonenfant C, Gaillard JM, Courchamp F. 2011b. Cat dilemma: Too Protected to
Escape Trophy Hunting? *PloS ONE* **6**, e22424. (doi:10.1371/journal.pone.0022424)

7. Darimont CT, Coddling BF, Hawkes K. 2017. Why men trophy hunt. *Biol. Lett.*, **13**,
20160909. (doi:10.1098/rsbl.2016.0909)
8. Darimont CT, Child KR. 2014. What Enables Size-Selective Trophy Hunting of
Wildlife? *PLoS ONE* **9**, e103487. (doi:10.1371/journal.pone.0103487)
9. Festa Bianchet M. 2003. Exploitative wildlife management as a selective pressure for life-
history evolution of large mammals. In *Animal behavior and wildlife conservation* (eds M
Festa Bianchet, M Apollonio), pp. 191–207. Washington, DC:Island Press.
10. Darwin C. 1874. *The Descent of Man and Selection in Relation to Sex*. 2nd ed.
London:John Murray.
11. Zahavi A. 1975. Mate selection—A selection for a handicap. *J. Theor. Biol.* **53**, 205–214.
(doi:10.1016/0022-5193(75)90111-3)
12. Grafen A. 1990. Biological signals as handicaps. *J. Theor. Biol.* **144**, 517–546.
(doi:10.1016/S0022-5193(05)80088-8)
13. Getty T. 1998. Handicap signalling: when fecundity and viability do not add up. *Anim.*
*Behav.* **56**, 127–130. (doi:10.1006/anbe.1998.0744)
14. von Rueden C. 2014. The roots and fruits of social status in small-scale human societies.
In *The psychology of Social Status*. pp. 179–200. New York, NY:Springer.
15. Bliège Bird R, Smith EA. 2005. Signaling Theory, Strategic Interaction, and Symbolic
Capital. *Curr. Anthropol.* **46**, 221–248. (doi:10.1086/427115)
16. Veblen T. 1899. *Theory of the leisure class: an economic study in the evolution of*
*institutions*. New York, NY:Macmillan.
17. Grose J. 2011. Modelling and the fall and rise of the handicap principle. *Biol. Philos.* **26**,
677–696. (doi:10.1007/s10539-011-9275-1)
18. Coddling BF, Bird RB, Bird DW. 2011. Provisioning offspring and others: risk–energy trade-
offs and gender differences in hunter–gatherer foraging strategies. *Proc. R. Soc. Lond. B.*
*Biol. Sci.* **278**, 2502–2509. (doi:10.1098/rspb.2010.2403)
19. Hawkes, K. 1991. Showing off: Tests of an hypothesis about men's foraging goals. *Ethol.*
*Sociobiol.* **12**, 29–54. (doi:10.1016/0162-3095(91)90011-E)
20. Bliège Bird R, Smith EA, Bird DW. 2001. The hunting handicap: costly signaling in human
foraging strategies. *Behav. Ecol. Sociobiol.* **50**, 9–19. (doi:10.1007/s002650100338)

- 21. Smith EA, Bird RB, Bird DW. 2003. The benefits of costly signaling: Meriam turtle
hunters. *Behav. Ecol.* **14**, 116–126. (doi:[10.1093/beheco/14.1.116](https://doi.org/10.1093/beheco/14.1.116))
- 22. Ripple WJ *et al.* 2014. Status and ecological effects of the world's largest
carnivores. *Science* **343**, 1241484. (doi:[10.1126/science.1241484](https://doi.org/10.1126/science.1241484))
- 23. Safari Club International. 2014. *Safari Club International Record Book: World Hunting*
*Award Field Journal* ([https://www.safariclub.org/sites/default/files/2018-](https://www.safariclub.org/sites/default/files/2018-10/world%20hunting%20award.pdf)
[10/world%20hunting%20award.pdf](https://www.safariclub.org/sites/default/files/2018-10/world%20hunting%20award.pdf))
- 24. Silva M, Downing JA. 1995. *CRC handbook of mammalian body masses*. Boca Raton: CRC
Press.
- 25. Krausman PR, Demarais S. 2000. *Ecology and Management of Large Mammals in North*
*America*. Upper Saddle River, NJ: Prentice Hall.
- 26. Safari Club International. 2008. *The Safari Club International Online Record Book of Trophy*
*Animals*. Tucson: Safari Club International.
- 27. NatureServe. 2018. *NatureServe Explorer: An online encyclopedia of life*. Version 7.0.
NatureServe, Arlington, VA, U.S.A. (<http://explorer.natureserve.org/>)
- 28. Akaike H. 1973. *Information theory and an extension of the maximum likelihood principle*.
*Second International Symposium on Information Theory*. pp. 267–281.
- 29. Burnham KP, Anderson DR. 2002. *Model Selection and Multimodel Inference: A Practical*
*Information Theoretic Approach*. New York, NY: Springer.
- 30. Bates D, Maechler M, Bolker B, Walker S. 2015. Fitting Linear Mixed-Effects Models
Using lme4. *J. Stat. Softw.* **67**, 1–48. (doi:[10.18637/jss.v067.i01](https://doi.org/10.18637/jss.v067.i01))
- 31. R Core Team. 2017. *R: A language and environment for statistical computing*. R Foundation
for Statistical Computing, Vienna, Austria.
- 32. Nash JC, Varadhan R. 2011. Unifying Optimization Algorithms to Aid Software System
Users: optimx for R. *J. Stat. Softw.* **43**, 1–14. (doi:[10.18637/jss.v043.i09](https://doi.org/10.18637/jss.v043.i09))
- 33. Freckleton RP. 2011. Dealing with collinearity in behavioural and ecological data: model
averaging and the problems of measurement error. *Behav. Ecol. Sociobiol.* **65**, 91–101.
(doi:[10.1007/s00265-010-1045-6](https://doi.org/10.1007/s00265-010-1045-6))
- 34. Galipaud M, Gillingham MA, David M, Dechaume-Moncharmont FX. 2014. Ecologists
overestimate the importance of predictor variables in model averaging: a plea for cautious
interpretations. *Methods. Ecol. Evol.* **5**, 983–991. (doi:[10.1111/2041-210X.12251](https://doi.org/10.1111/2041-210X.12251))

35. Cade BS. 2015. Model averaging and muddled multimodel inferences. *Ecology* **96**, 2370-
2382. (doi:10.1890/14-1639.1)

36. Darimont CT, Fox CH, Bryan HM, Reimchen TE. 2015. The unique ecology of human
predators. *Science* **349**, 858-860. (doi:10.1126/science.aac4249)

37. Hall RJ, Milner-Gulland EJ, Courchamp F. 2008. Endangering the endangered: The effects
of perceived rarity on species exploitation. *Conserv. Lett.* **1**, 75-81. (doi:10.1111/j.1755-
263X.2008.00013.x)

38. Child KR, Darimont CT. 2015. Hunting for trophies: Online Hunting Photographs Reveal
Achievement Satisfaction with Large and Dangerous Prey. *Hum. Dimens. Wildl.* **20**, 531-
541. (doi:10.1080/10871209.2015.1046533)

39. Ebeling-Schuld AM, Darimont CT. 2017. Online hunting forums identify achievement as
prominent among multiple satisfactions. *Wildl. Soc. Bull.* **41**, 523-529.
(doi:10.1002/wsb.796)

40. Barnosky AD, Koch PL, Feranec RS, Wing SL, Shabel AB. 2004. Assessing the Causes of
Late Pleistocene Extinctions on the Continents. *Science* **306**, 70-75.
(doi:10.1126/science.1101476)

41. Ray J, Redford KH, Steneck R, Berger J. (eds.) 2005. *Large carnivores and the conservation*
*of biodiversity*. Washington DC: Island Press.

42. Estes JA *et al.* 2011. Trophic Downgrading of Planet Earth. *Science* **333**, 301-306.
(doi:10.1126/science.1205106)

 **Figure 1.** Effect of mass on daily guided-hunt price for carnivore (orange) and ungulate
 (blue) species in North America. Points show raw mass for carnivores and ungulates,
 curves show predicted means from the maximum-parsimony model (see text), and
 shading indicates 95% confidence intervals for model-predicted means.

Table 1. North American “big game” species included in our study.

Species (common)	Latin	Classification
Mountain Lion	Puma concolor	Carnivore
Black Bear	Ursus americanus	Carnivore
Brown Bear	Ursus arctos	Carnivore
Polar Bear	Ursus maritimus	Carnivore
Muskox	Ovibos moschatus	Ungulate
Gray Wolf	Canis lupus	Carnivore
Thinhorn Sheep	Ovis dalli	Ungulate
Bighorn Sheep	Ovis canadensis	Ungulate
Caribou	Rangifer tarandus	Ungulate
Pronghorn	Antilocapra americana	Ungulate
White-tailed Deer	Odocoileus virginianus	Ungulate
Moose	Alces alces	Ungulate
Mule Deer	Odocoileus hemionus	Ungulate
Mountain Goat	Oreamnos americanus	Ungulate
Elk	Cervus canadensis	Ungulate

**Table 2.** AIC evaluation of models for predicting prices of hunts offered by hunting
 guides. The explanatory variables were the average male body mass (Mass; in kg), the
 classification (Class; either carnivore or ungulate), descriptions of difficulty or danger in
 Safari Club International hunt descriptions (SCI; either absence or presence), and
 conservation status in jurisdiction (Status; 1, 1.5, 2, ..., 5).

Model	df	logLik	Δ AIC	AICw
Mass + Class + Mass×Class	7	-48778.6844	0.00	0.23
Mass	5	-4880.2597	1.132	0.13
Class + Mass + SCI+ Mass×Class	8	-48778.4547	1.543	0.11
Class + Mass + Status + Mass×Class	8	-48778.520	1.6458	0.10
SCI + Mass + SCI×Mass	7	-48789.7144	2.056	0.08
Status + Mass	6	-4880.0978	2.8174	0.06
Class + Mass + SCI + Mass×Class + SCI×Mass	9	-4877.1487	2.923	0.05
SCI + Mass	6	-4880.1688	2.954	0.05
Null (intercept only)	4	-48823.302	3.272	0.05
SCI + Mass + Status + SCI×Mass	8	-48789.5930	3.8278	0.04
Status + Class + SCI + Mass + SCI×Mass + Mass×Class	10	-4877.070	4.6459	0.02
SCI + Mass + Status	7	-4880.0174	4.679	0.02
Status	5	-4882.814	4.9284	0.02
SCI	5	-4882.2999	5.2348	0.02
SCI + Status	6	-4882.1380	6.9078	0.01

984
 985

Appendix B

August 12, 2019

Dear Royal Society Open Science team,

Thank you for the opportunity to submit our revised manuscript "**Trophy hunters pay more to target larger-bodied carnivores**" RSOS-191231.R1, following your request for minor revisions.

We have addressed Reviewer 2's comments and modified our manuscript's title accordingly. We identify these changes below by using ** before our response. New or revised text are in italics.

We also upload a Track Changes version of the manuscript indicating these changes.

Sincerely,

Ilona Mihalik,
on behalf of Andrew Bateman and Chris Darimont

Associate Editor Comments to Author (Dr Claudia Wascher):

The authors have significantly revised the manuscript alongside with previous comments by the reviewers. The reviewers recommend the manuscript to be accepted for publication, they only ask the subtitle to be deleted.

**** Thank you. We have made these changes.**

Reviewer comments to Author:

Reviewer: 1

Comments to the Author(s)

The changes addressed my concerns- looks good.

**** Thank you.**

Reviewer: 2

Comments to the Author(s)

The authors have taken the reviewers' comments to heart and greatly clarified their manuscript. The distinction between "cost" and "price" is especially helpful.

The only change I would recommend is the deletion of the subtitle. It does not seem necessary. The title alone summarizes the manuscript's main finding.

**** Thank you for your suggestion. We have removed the subtitle and our title now reads,**

Trophy hunters pay more to target larger-bodied carnivores